# The Health Literacy Status and Its Role in Interventions in Iran: A Systematic and Meta-Analysis

**DOI:** 10.3390/ijerph18084260

**Published:** 2021-04-17

**Authors:** Seyedeh Belin Tavakoly Sany, Hassan Doosti, Mehrsadat Mahdizadeh, Arezoo Orooji, Nooshin Peyman

**Affiliations:** 1Department of Health Education and Health Promotion, Faculty of Health, Mashhad University of Medical Sciences, Mashhad 13131-99137, Iran; tavakkolisanib@mums.ac.ir (S.B.T.S.); mahdizadehtm@mums.ac.ir (M.M.); 2Social Determinants of Health Research Center, Mashhad University of Medical Sciences, Mashhad 13131-99137, Iran; arezoo.orooji23@gmail.com; 3Department of Mathematics and Statistics, Macquarie University, Sydney, NSW 2109, Australia; hassan.doosti@mq.edu.au; 4Department of Epidemiology and Biostatistics, Faculty of Health Sciences, Mashhad University of Medical Sciences, Mashhad 13131-99137, Iran

**Keywords:** health literacy, health literacy interventions, health-promoting behaviors, meta-analysis, self-efficacy, systematic review

## Abstract

There are increasing calls for public health policies to realize the visions of a health literate society and health literacy on a global scale. However, there are still more gaps in what researchers recognize and what steps they should take to improve health literacy (HL) skills. This review aimed to measure the HL status of the Iranian population and the effect size of the underlying association between HL and other health outcomes, and to examine the effectiveness of HL interventions on improving the functional dimension of HL, self-efficacy, and health-promoting behaviors. All full text published articles written in English and Persian language were included from inception until January 2019, but the type of study is not limited. A total of 52 potentially relevant articles with data on 36,523 participants were included in this review. In the population with health conditions, the average HL score was 62.51 (95% CI: 59.95–65.08), while in the patient population, the HL score was 64.04 (95% CI: 60.64–67.45). Health literacy was positively and significantly correlated with self-care behaviors 0.42 (95% CI; 0.35–0.49), self-efficacy 0.35 (95% CI; 0.26–0.43), knowledge 0.50 (95% CI; 0.44–0.55), communication skills 0.33 (95% CI; 0.25–0.41), and health promotion behaviors 0.39 (95% CI; 0.35–0.44). The meta-analyses showed that overall, HL interventions significantly improved HL status, self-efficacy, and health promotion behaviors. Results indicate that HL status was in the range of marginal HL level in the Iranian population. Our finding highlights the beneficial impact of HL intervention on health-promoting behaviors and self-efficacy, particularly in low literacy/socioeconomic status people.

## 1. Introduction

Health literacy (HL) is a key element of health promotion that has been increasingly investigated since the 1990s. This term was first published in 1974 at the health education conference, which discussed the determinants of social health that affect mass communication, the health-care system, and health education issues [1]. Relevant evidence had shown that HL is a complex issue, and low HL can increase an individual’s health status, health outcomes, and the risk of misinterpreting treatment instructions [2,3]. The World Health Organization (WHO) defines HL as follows: “health literacy implies the achievement of a level of knowledge, personal skills and confidence to take action to improve personal and community health by changing personal lifestyles and living conditions”. This definition confirms that the concept of HL is not simply a functional ability (the ability to read and write), it includes different skills that empower people to attend more fully in community and to exert a higher degree of control over their health action and health decision-making [4].

The concept of HL has emerged from two distinctive roots that describe HL, respectively, as a personal ‘‘asset’’, or a clinical ‘‘risk’’. The clinical ‘‘risk’’ reflects recognition of the impact that low HL can have on the effectiveness of health service organization and clinical care. Improved sensitivity of service organization and clinicians can enhance the quality of patient–health provider communication and improve access to health care services. This leads to a health service organization better placed to provide patient education and management that will help to improve patient ability and health outcomes associated with successfully adhered to recommended clinical care [2,4]. The concept of HL as an “asset” has evolved from roots in public health, health promotion, health education and communication. This conceptualization is seen as a means to developing individuals’ skills and ability to exert greater control over their health action (social, personal, and environmental) and health decision-making. Actions to improve HL are focused on the development of context intended to promote individual’s health knowledge, self-efficacy, and self- management. This conceptualization of HL is less well tested through systematic research. Research to support the ‘‘asset’’ model is at a basic stage and it remains the main idea. Likewise, if achieving HL as described by WHO is to be the main aim, different measurement tools will be needed for different stages and ages. Different measurement tools will be required to distinguish between interactive, critical, and functional HL. Although assessing the concept of HL is not a totally new challenge in the social sciences, there will need to be comprehensive testing to ensure that the different measures not only focus on health information for personal benefit, but also on the social determinants of health [4].

Globally, HL policies and strategies are not yet well-known among health decision-makers and politicians. According to WHO, today’s reaction-based health care centers are not suitable for the challenges of the recent century because half the world’s population has access to poor quality health care services. Health systems’ transformation around the needs of communities and people are more effective at improving HL and patient engagement [3]. Therefore, in the recent decade, great attention has been paid to improve individuals’ HL skills (including reading, writing, listening, speaking, numeracy, and critical analysis) in the different communities [5,6]. Likewise, for 40 years, HL intervention has been used as a key educational program to promote individual’s health status. Several studies have shown that health education interventions are incompatible with the characteristics of different populations. Previous research emphasized that HL interventions should be culturally adapted, evidence-based, and conducted by health professionals or providers [6,7]. From the more narrow perspective of the health system, the effectiveness and implementation of HL intervention are still equivocal in different communities because of methodological limitations and changes in health action [5,7,8]. According to the available evidence, it is feasible to assess the HL status in different communities to clarify and understand the impact of HL on health inequality and health status [9,10].

In Iran, the term of HL was first entered in 1974 as a key element of health promotion, and it has been increasingly investigated since the 1990s [9,10,11,12,13,14,15,16,17,18,19,20,21,22,23,24,25,26,27,28,29,30,31,32,33]. To the best of our knowledge, no conceptual model for HL has been implemented in Iran including both a public health perspective and medical assistance. There is no rigorous national assessment of the overall status of HL in the Iranian population [34,35]. It is unclear which personal and psychological characteristics will influence HL intervention, and which pathways link Iranian patients with their healthcare providers and health outcomes. Given the importance of HL to public health, it is wise to examine what is being done to evaluate HL in the Iranian population for the first time. This effort will be practical in evaluating whether the health and social ambitions of Iran for sustainable health-promotion are well achieved. Therefore, we aim (1) to systematically review the status of HL in Iranian population, and the association between HL and its influencing factors; and (2) to examine whether HL interventions targeting the Iranian population can improve HL status, health promoting behaviors, and self-efficacy. Further, the challenges, gaps, limitations and the need for future research are discussed.

## 2. Materials and Methods

We planned a systematic review and meta-analysis based on the Cochrane collaboration tool [36] and the preferred reporting items for systematic reviews and meta-analyses (PRISMA) [37]. We addressed the following research questions using review techniques:What is the HL status of the Iranian population?What are the commonly used instruments for measuring HL in Iran?Is there an association between individuals’ HL and self-care behaviors, self-efficacy, health promotion behaviors (healthy food and physical activity), medical adherence, knowledge, and communication skills?Can HL intervention improve HL skills (includes reading, writing, listening, speaking, numeracy, and critical analysis) and health outcomes?

### 2.1. Search Strategy

In this review, a total of seven electronic databases were searched, including the Cochrane database, PubMed, MEDLINE, Scopus, Scientific Information Database (SID), Web of Knowledge, Google scholar, and Educational Resource Information Center (ERIC), from inception until January 2019. Furthermore, we hand-searched the reference list of all studies to find additional studies that were ignored by the search terms. We used the Medical subject heading (MeSH) keywords related to the term “health literacy” and combined with the following terms: “educational intervention”, “literacy”, “health providers”, “medication adherence”, “communication skills”, “self-efficacy”, “knowledge”, “health promotion behaviors”, “self-care behaviors”, and “primary care”.

### 2.2. Article Selections and Screening

All details about the inclusion and exclusion criteria are summarized in Appendix A. Two authors conducted a preliminary search and independently screened the titles, abstracts, and full texts of retrieved articles. Then, we selected articles meeting the inclusion criteria. Both reviewers were in 100% agreement over all included articles and extracted data. Furthermore, the third author resolved any doubts and discrepancies regarding the included articles. Health literacy interventions are defined as all educational interventions designed based on HL strategies, aiming to increase people’s engagement in health promotion behaviors and health decision-making, and improve HL skills.

### 2.3. Data Extraction and Quality Assessment

We used a pre-designed form adapted for a meta-analysis and systematic review [18] to conduct independent double data extraction and the following information was extracted: (1) the publication year/author(s), type of publication, city, study design, aims, participant characteristics, and sample size; (2) HL and health outcomes scores; (3) methodological approach, setting, and type of instruments used to measure HL and other health outcomes; (4) estimation of the association between HL and health outcomes with corresponding *p* values, the statistical test; and (5) type of intervention, duration, method analysis, number of participants, control and intervention condition, follow-up, and effect size. Full-text articles were retrieved for eligibility assessment based on inclusion and exclusion criteria. Then, an independent dual rating was performed to consider and appraise the quality of the included studies. Authors resolved all disagreements and doubts through discussion. The quality of included studies was assessed using the Cochrane risk of bias tool [36], Downs and Black quality assessment scores [38], and JBI critical appraisal checklist [39] to examine the quality of randomized controlled trials (RCT), non-randomized intervention, and cross-sectional studies, respectively.

### 2.4. Meta-Analysis Assessment 

We included all quantitative information for the meta-analysis if they had a validated measure of HL and were directly related to HL. In this study, HL interventions were included as a primary target in the meta-analysis. The effect size (ES) of HL status was measured based on the mean and standard deviation as the unit of analysis. For the effect sizes of relationships and intervention, we included several formats across studies such as *t-*test values, linear-regression coefficients, correlation coefficients, and χ^2^ values. Since the correlation coefficient (*r*) is the most common effect size in the meta-analysis program, and represents both the direction and the magnitude of the relationship, therefore, all these effect sizes were transferred to the effect size measure—correlation. A negative “*r”* shows that HL is inversely associated with better health outcomes; a positive “*r*” indicates that HL is associated. For the intervention studies, a positive “*r*” indicates that the HL intervention program can effectively improve HL skills and health promotion behaviors, while a negative “*r*” indicates that the HL intervention is ineffective.

A meta-analysis was conducted using a fixed-effects model to assess the heterogeneity across the effect sizes with 95% confidence intervals. The random-effects model is also used in this study, as the true association between HL and outcomes. Heterogeneity was estimated using Thompson’s *I*^2^ statistic and *Q* value. The *I*^2^ is a measure of the ratio of the total variation in study measures due to heterogeneity. The *Q* value examines whether the invariance of effects is not greater than the invariance caused by the sampling error. If the *Q* value corresponds with *p* < 0.05, it devastates a random-effects model and heterogeneity is appropriate for interpretation [36]. In this study, we first ran fixed-effects models and the heterogeneity tests, and then ran the random effects model. Likewise, to better test our hypothesis, the meta-analysis was stratified by HL instruments and different health outcomes. For the HL intervention, all studies identified results of an independent group, and all quantitative results were included if they are directly relevant to HL. In this study, only HL intervention for the Iranian population was used for the meta-analysis. We calculate the effect size of the intervention study based on the standardized mean difference (SMD) [7,40]. Meta-regression analysis was conducted to examine the influence of covariates on the estimation of pooled effects, and to assess which study characteristics are feasible for stratified analysis. The potential publication bias was investigated using funnel plots. The significant differences were considered as *p* < 0.05. We also used STATA software, version 14 (Corp LP) to analyze all statistical tests. A systematic review was performed in parallel.

## 3. Results

### 3.1. Search Outcome

A total of 52 potentially relevant articles with data on 36,523 participants were included in the systematic review. Of these, 47 studies with data on 34,171 participants were included in the meta-analysis (Figure 1, Table 1). All data were collected from 30 different cities in Iran, and 48% of the research was conducted in Tehran, Mashhad, and Esfahan. The sample size was ranged from 43 to 20,571. We noticed that the first studies were published in 2011 and 2012, and 73% of the articles were published in the past five years (Table 1).

### 3.2. Study Designs and Populations

Of 52 included studies, 25 (48%) studies included populations with health conditions [9,10,11,12,13,14,15,16,17,18,19,20,21,22,23,24,25,26,27,28,29,30,31,32,33], and 24 (46.3%) studies included patient participants with chronic disease [12,27,41,42,43,44,45,46,47,48,49,50,51,52,53,54,55,56,57,58,59,60,61,62], and other disease (3/52; 5.7%) [55,56,59] (Table 1). We included 45 cross-sectional studies [9,10,11,12,13,14,15,16,17,18,19,20,21,22,23,24,25,26,27,28,29,30,31,32,33], 3 randomized control trials (RCT) [42,43,51], and 4 semi-experimental studies [20,25,52,62] evaluated 7 interventions ranging from education sessions or coaching sessions targeted at patients or health providers, booklets or DVD decision aids and counseling sessions. The development and design of six out [20,25,42,51,52,62] of seven interventions are developed and designed specifically for low socioeconomic (literacy and low income) people with chronic diseases. All these studies used the health literacy framework to improve HL skills, self-efficacy, and health promotion behaviors of the intervention program.

### 3.3. Methodological Quality

Full-text articles were retrieved for eligibility assessment based on inclusion and exclusion criteria. In this study, 52 studies met the inclusion criteria, which included 45 cross-sectional studies [9,10,11,12,13,14,15,16,17,18,19,20,21,22,23,24,25,26,27,28,29,30,31,32,33], 3 randomized control trials (RCT) [42,43,51], and 4 semi-experimental studies [20,25,52,62]. The quality of included studies was low, across RCTs. Three studies were rated according to the Cochrane risk of bias tool. All RCTs did not clearly report allocation concealment, the blinding of participants, and the blinding of outcome assessment while all included studies reported the random sequence generation, selective reporting, outcome assessment, and other sources of bias. The quality of included semi-experimental studies was acceptable with the average score of 13.6. It ranged from 12.9 to 14 out of 26. This is moderate quality but consistent with Downs and Black quality assessment scores for non-randomized intervention [38]. The quality of included studies was moderate, across cross-sectional studies. All cross-sectional studies were rated against the JBI critical appraisal checklist (Appendix A). Almost all of the studies clearly defined criteria for inclusion, study subject, and the setting of appropriate statistical analysis. All cross-sectional studies also used objective, standard criteria for measurement of the condition and provided a valid and reliable way for outcome measurement. Five studies did not identify the confounding factors and strategies to deal with confounding factors. Only three studies used the valid and reliable way to measure exposure.

### 3.4. Health Literacy Tools

In this review, 31 (59.6%) studies measured HL using the Test of Functional Health Literacy in Adults (TOFHLA) [9,10,15,16,17,18,19,20,21,22,23,27,41,42,43,46,47,49,50,51,52,53,54,55,56,57,62], and 6 (11.5%) studies used the Short Test of Functional Health Literacy in Adults (S-TOFHLA) [19,25,56,57] in the culturally adapted version. Eleven studies (21%) assessed HL using the Health Literacy for Iranian Adults (HELIA) in the original version. Three studies assessed HL using the Newest Vital Sign (NVS) (Table 1 and Appendix A). Existing measures of HL, such as S-TOFHLA, TOFHLA, and HELIA are designed based on comprehension of general health information, functional skills, and decision-making [20,25,42,43,51,52,62].

### 3.5. Systematic Review

Our findings showed that TOFHLA and HELIA were the most common questionnaires used to measure HL, while other instruments were rarely used (Table 1). Most of the studies on HL have been conducted among Iranian adults (male and female) aged between 18 and 60 years, but there are three studies focused on measuring the HL level of the elderly population [16,22,64]. Furthermore, most of the HL studies were also conducted in low literacy/low socioeconomic populations in Iran, and only three studies examined the HL status among groups with higher socioeconomic status [15,33,43]. As shown in Table 1, 60.8% of total participants were illiterate or under diploma, and 64% of participants were low or moderate-income. Of the 28 studies that assessed association between socio-demographic characteristics and HL, 28 studies reported that HL status is significantly (*p* < 0.05) associated with education level and family income [10,11,14,16,18,20,21,23,24,25,26,27,33,41,45,49,50,51,52,53,54,57,58,59,60,61], and 20 studies reported significant inverse correlation (*p* < 0.05) between age and HL [18,21,23,24,25,26,27,33,41,45,49,50,51,52,53,54,57,59,60,61,62]. These associations were reported more frequently in other studies [6,66]. However, these results were significantly heterogeneous, so it is not possible to determine trends or draw conclusions about the overall impact of socio-demographic characteristics on HL level. A total of 36,523 participants were included in this systematic review, of which 14,364 participants (39.32%) had inadequate HL and 60.66% of total participants showed marginal (10,787, 29.53%) and adequate HL (11,369, 31.13%). Seven studies examined the intervention’s effect based on HL strategies on HL skills, self-efficacy, and health promotion behaviors (physical activity, medication adherence, and healthy diet) in patients with chronic disease. These studies showed significant increases of HL skills, self-efficacy, and health promotion behaviors in the intervention group compared to the control group [20,25,42,51,52,62]. Likewise, there was a statistically significant effect of the educational intervention on condition-specific health outcomes [51,52,62].

### 3.6. Meta-Analysis

#### 3.6.1. Health Literacy Status

Of the 52 studies included in systematic review, 47 studies met the inclusion criteria for the meta-analysis to assess HL status. Given the different instruments that were used to assess the HL, we stratified data based on the common instruments to increase the power of meta-analysis to detect the impact of instruments on the overall effect size (ES) of HL.

The total score of included studies gave an overall ES of 62.51 (95% CI: 59.95–65.08) for the average of HL among participants with a health condition, suggesting that HL status is in the range of marginal literacy level in Iran. The reported pooled effect of TOFHLA, S-TOFHLA, and HELIA is estimated to be 63.47 (95% CI: 60.06–66.88), 60.28 (95% CI: 56.28–64.31), and 76.26 (95% CI; 60.47–92.04), respectively. The analysis showed that the overall ES of HL derived from the studies conducted with the HELIA instrument was significantly greater than the studies conducted with TOFHLA/STOHFLA. The I^2^ statistic was not significant (10.5%, *p* = 320) for overall ES of HL, and it was 23.9% (*p* = 0.202), 0.0% (*p* = 0.460), and 0.0% (*p* = 0.884) from studies using TOFHLA, S-TOFHLA, and HELIA, respectively, suggesting a suitable homogeneity within the fixed-effects results. The analysis revealed that use of a different study design, study setting, age, education, income, time of study and gender did not have a significant effect on the overall ES of HL across any of the instruments (Figure 2).

Figure 3 and Figure 4 show results of meta-analysis in the patient population. Meta-regression analysis revealed that the study time was a significant covariate of HL, and has an effect on the overall ES (*p* < 0.05) of HL among the patient population (Appendix A). According to the individual analysis, studies that were conducted from 2012 to 2014 had a significant outlier compared to other studies (Appendix A). These outliers may decrease the pooled effect size of HL in patient participants. Therefore, we analyzed the HL status based on two time series in the patient-population. This helped us to increase the power of analysis to detect the impact of study time on the overall ES estimate. This indicated that the overall ES of the studies that were conducted from 2012 to 2014 was 40.15 (95% CI; 37.50–42.81), suggesting that the HL status of patient participants before 2015 was in the range of inadequate HL levels (Figure 3). In the case of studies that were conducted after 2014, the overall ES of HL was 64.04 (95% CI: 60.64–67.45), suggesting that HL status is in the range of marginal HL level in this population. The *I*^2^ statistic for overall ES of HL was 0.0% (*p* = 0.996), and it was 0.0% (*p* = 0.998), 0.0% (*p* = 0.968), and 8.6% (*p* = 0.357) from studies using TOFHLA, S-TOFHLA, and HELIA, respectively. These findings suggested that a fixed-effects result was suitable for interpretation (Figure 4).

#### 3.6.2. Binary Outcome

Across 15 studies, using fixed-effect models, HL was positively and significantly correlated with self-care behaviors 0.42 (95% CI; 0.35–0.49) [11,23,42,49,52,56,60], self-efficacy 0.35 (95% CI; 0.26–0.43) [49,51,52], knowledge 0.50 (95% CI; 0.44–0.55) [42,44,46,52], communication skills 0.33 (95% CI; 0.25–0.41) [46,51], and health promotion behaviors 0.39 (95% CI; 0.35–0.44) [11,20,26,42,49] (Figure 5). It is obvious from the forest plot that the correlation measure from studies between HL and knowledge was larger than other outcomes. The *I*^2^ statistic (50.08%) was also significant (*p* = 0.047) for the self-care behaviors, suggesting that a random-effects model was suitable for interpretation. This was likely due to Sharafi and Hejazi’s studies [30,52], which largely over-inflating the pooled effect estimates for the association between HL score and self-care behaviors. The Egger’s test showed an insignificant (*p* = 0.964) publication bias.

#### 3.6.3. Intervention Outcome

Overall, the meta-analysis for HL intervention studies shows a positive significant effect across the domains (HL skills, self-efficacy, and health promotion behaviors), indicating that the HL interventions increase people engagement in health promotion behaviors and health decision-making, and improve HL skills [25,43,51,52,62]. However, the pooled effect of HL intervention on self-efficacy was not significant. The reported pooled effect estimates for HL skills, self-efficacy, and health promotion behaviors were 15.53 (95% CI; 6.82–24.24), 10.19 (95% CI; −3.23–23.61), and 15.30 (95% CI; 2.22–28.37), respectively (Figure 6). The *I*^2^ was not significant for the HL skills, self-efficacy, and health promotion behaviors, indicating that a fixed effects model was suitable for interpretation for all these domains. However, it is possible that the meta-analysis lacked sufficient power to assess the effect of HL intervention on each domain because of the low number of studies for each domain. Individual studies showed that Tavakkoli’s ES [51,62] for HL skills were a significant outlier compared to the other studies. A sensitivity analysis revealed that the overall ES for the HL domain is 4.67 with the omission of Tavakkoli’s surveys. This result provided evidence that Tavakkoli’s studies on patient population might increase the overall ES of both the HL domain and the intervention-designed studies. Investigation of the funnel plot also indicated the large deviation that Tavakkoli’s study on HL skills of hypertensive patients was having relative to the other studies. This plot shows no explicit gaps and *n* individual studies at both the low and high end of the range. Therefore, the symmetry of the plot was reasonable (Figure 7).

## 4. Discussion

### 4.1. Health Literacy Status

One of the main objectives of this review was to identify the HL status in the Iranian population, based on existing literature and evidence. As shown in the systematic review, most of the participants had inadequate or marginal HL. Therefore, HL is a growing national concern in Iran, like results from some national HL surveys that were conducted in other countries [6,66,67]. Limited HL (inadequate plus marginal) is prevalent in many communities in both developed and developing countries, accounting for one-third to one-quarter of the population. According to the 2009 National Assessment of HL in the United States of America (USA), it is estimated that nearly 12% of adults in the USA have adequate HL levels, 53% have marginal HL levels, and 36% have insufficient or basic levels [67,68]. A recent study that was conducted in the eight European countries on adult HL skills has shown that approximately 56% of men and women have inadequate HL levels (21%) or marginal (35%) [69], and limited HL was ranged from 29% in the Netherlands and 62% in Bulgaria [69]. Similarly, other studies have estimated the HL status among the Latino women in Philadelphia. They reported that up to 50% of the women have inadequate HL levels [70]. In Australia, the national HL survey conducted in 2006 showed that up to 52% of women and 57% of men in Australia have basic HL levels lower than the level required to meet complex needs of daily work and life [71,72].

The meta-analysis revealed that the overall HL status for included studies conducted during 2012 to 2014 was in the range of inadequate level, while there has been a significant improvement in the overall trend of HL status to marginal level after 2015. This may be due to the recent improvement in the epidemiological characteristics of Iran’s healthcare characteristics [73,74]. In 2014, a stepwise plan, called the Health Sector Evolution Plan (HSEP), was launched in the healthy development national strategies in Iran. HSEP included multiple interventions and series of reforms to improve quality of hospital care and access to healthcare such as providing free basic health insurance to all Iranians, promoting primary care quality in health centers and hospitals, reducing out-of-pocket (OOP) payments for inpatient services, developing policies to encourage medical doctors to stay in deprived areas, updating tariffs to more realistic values, financial protection of patients with specific diseases or chronic disability, and promoting the family physician program and health services [75,76]. Recent studies reported that the HSEP program in Iran reduces the incidence of diseases and increases healthy lifestyles, health promotion behaviors, and the quality of the health care system [77,78]. In Iran, although the epidemiologic profile has been significantly changed, inadequate HL of people and self-care behaviors particularly in the low socio-economic population are the main concern for policy makers [45,78]. The inadequate HL can potentially threaten the efficiency and sustainability of HSEP because it is associated with a poor understanding of health information self-care management skill, and adherence to treatment leading to poor health outcomes such as an increase in the medical costs, hospitalizations, and higher mortality [78,79].

Likewise, the systematic review shows that TOFHLA and HELIA are the most common questionnaires used to measure HL. This may be due to the common usage of these questionnaires, representativeness of health-related duties, and their association with fluid cognitive abilities [80,81]. In addition, based on socio-demographic characteristics (education level, age, and income), differences in HL levels were found. Other studies report these differences more frequently [5,26,82]. A general trend showed that individuals with lower family income and educational levels have lower levels of HL and self-managed skills to engage health promotion behaviors, and have greater difficulties in understanding health information. However, the limited sample size and heterogeneity of studies using socio-demographic characteristics as a predictor or proxies does not permit us to conclude the effect of people’s socio-demographic characteristics on the level of HL skills. Caution is needed when interpreting these results, and these data were not used in the meta-analysis process. Therefore, it is essential to conduct longitudinal studies to assess the effect of socio-demographic characteristics on HL status.

In addition, the limited number of studies examines the association between people’s HL and health promotion behaviors or health information. However, the meta-analysis showed that HL has a positive significant association with people’s self-care behaviors, self-efficacy, knowledge, communication skills, and health promotion behaviors. On the other hand, the included studies give us clear evidence that adequate HL led to an improvement in these outcomes [11,23,42,49,52,56,60]. Indeed, people with adequate HL use health skills that correspond to the well-established health information and behaviors [26,79].

### 4.2. Health Literacy Intervention

One of the main research questions of this review was to collect evidence on the effect of HL intervention on the improvement of HL skills, self-efficacy and health promotion behaviors. The studies included in the meta-analysis provide the clear evidence that HL interventions play an important role in improving these domains [25,43,51,52,62]. This result has been confirmed in more studies, which show that good HL intervention can greatly improve the understanding of health information and activation levels [6,72,83]. Given the diversity of designs of the included studies and limited sample size for each domain, these results need to be interpreted with caution and bear in mind the effect of Tavakkoli’s study as a significant outlier [51,62]. While the inclusion of the Tavakkoli studies showed a large effect of HL intervention on improving patient HL skills, we think that the overall effect measured without the Tavakkoli studies shows a much more precise effect estimate, which certainly fits the trend observed in the other outcome.

The included studies in the systematic review showed that people’s HL skills, self-efficacy, and health promotion behaviors significantly increase in the intervention group compared to the control group, particularly in low-socioeconomic participants [25,43,51,52,62]. These studies did not compare low socioeconomic patients to privileged patients. However, they demonstrated that although a disparity was higher among low socioeconomic participants in pre-intervention, disparities disappeared after the intervention [43,51,62].

They also highlighted that despite lower literacy and knowledge levels in low socioeconomic groups, they had more intention to understand the intervention’s content, even when the intervention was adapted at mixed literacy groups [51,52,62]. This suggests that educational interventions based on HL strategies could be more beneficial for the patients in low socioeconomic groups, and could in turn district health disparities in treatment preferences, decisional conflict, knowledge, and uncertainty. All included studies in the systematic review were not unique to low socioeconomic participants, but consistent with the results of many studies in other communities, which found a significant effect of HL intervention in patients on the improvement of health information and condition-specific health outcomes [6,72,83]. Likewise, a study in this review examined the impact of interventions based on HL strategies on patient–provider communication skills and decisional conflict in hypertensive patients [51]. This study reported that brief communication skills training based on HL strategies for health providers may be an efficient way to improve hypertension outcome. Furthermore, three studies highlighted that clear content, format, length of the intervention, simple language, and clarity interfered with the intervention’s effect [51,52,62].

### 4.3. Challenge and Gaps

In this study, we tried to highlight some of the major gaps and challenges in the field of HL research. The first gap in current research is that most HL studies were conducted in the general population with mixed socio-economic status. However, more than 60.8% of the total participants in this review were illiterate or under diploma, and 64% were low or moderate-income. The finding in the current review does not allow us to compare people’s HL skills in different socioeconomic levels. Therefore, further research should be conducted to specifically investigate the effectiveness of HL instruments and intervention on low socioeconomic patients or privileged patients.

The second gap highlighted the lack of a fixed and clear definition for HL because some health professionals define more issues beyond the framework of health information [6,79]. This problem has limited collaboration works and international research on the HL concept.

A third main gap is that the primary healthcare centers and hospitals in Iran have not been managed based on their patients’ literacy and HL levels. This may be a result of the disagreement in the definition of HL that affects how it can be categorized and estimated [77,78]. According to the study that was conducted on Iranian health providers in 2017, healthcare providers have low effective communication skills to guide their patients who have low literacy to understand and read all types of health-related materials. Likewise, most of the providers are unaware of the magnitude of this problem [46,51]. In reality, most patients in Iran have difficulty communicating with their physicians or providers and following up with medication instructions due to poor health knowledge, limited understanding of basic health vocabulary, and trouble in interpreting new concepts and information [46,51,77]. In Iran, although HL is fundamental for patient’s understanding of health information, participation in treatment options, informed decisions, and adherence to appropriate treatment, it continues to lack systematic attention from healthcare systems and medical education [74,75]. Therefore, there is an urgent need to implement a training program in Iran clinician systems to improve provider–patient communication skills and advise them for improving communication with patients.

Fourth, no locally comprehensive screening instrument was found to categorize and estimate HL. Our finding in this review confirms this problem. As shown in this review, different instruments are frequently used to measure HL, and TOFHLA and HE-LIA (local tool) were the most common questionnaires. Further, the result of meta-analysis revealed that the overall ES of HL from studies using the HELIA instrument was markedly higher than those using TOFHLA or STOHFLA. This may be a result of the difference in the content of the questionnaire, examining items, and cutoff scores to defining the level of HL. The HELIA is a locally validated instrument in Iran to examine psychometric parameters based on the 5-point Likert scale but it is less sensitive for evaluating well-functioning HL skills [84,85]. This may have increased the likelihood of more measurement challenges and made conflicting results. It is interesting to note that there is no gold standard for estimating HL as there is no one clear definition of HL [6,66,79]. Several studies indicated that HL concepts need to be understood and designed based on the background of the local community [67,71], like the HL instruments that were designed in Japan [67], the USA, and Australia [71]. This may be due to the fact that the HL concept is culturally specific in different communities and there are difficulties to reconcile the differences in education and culture systems [8,71]. In this context, further studies should be conducted to design a comprehensive validated screening instrument for HL based on its own local predictive factors. With a validated local reference tool, healthcare providers know what exactly can be managed, and can be measured [8,79].

Last, the qualities of the included studies in interventional studies were fairly low and variable. This is primarily due to the small sample size, poor sampling design, limiting systematic follow-up, and a lack of longitudinal interventional studies. This may have increased the likelihood of different biases that reduce the quality of intervention and health inequalities in routine clinical care.

### 4.4. Limitations

Given the paucity of study in the HL concept, we include all study designs in this review. Likewise, there are no longitudinal studies on HL status in the Iranian population. Therefore, the low sample size increased heterogeneity to estimate the association between the HL level and socio-demographic parameters. However, this introduced association with significant heterogeneity was not pooled in the meta-analysis and only inferred in systematic reviews. We used a fixed-effects model and a random-effects model to estimate the effect of the heterogeneity. This analysis suggested a suitable homogeneity within the fixed/random-effects results. Meta-regression analysis revealed that the study time was a significant covariate of HL, and has a significant impact on the overall ES (*p*< 0.05) of HL among the patient population.

The second limitation is the low quality of the interventional studies that have further reduced the number of included intervention studies. However, this finding is consistent with the scores of quality assessments published in the Cochrane review of Decision Aids and confirms the need for improvement in the methodology of interventional studies that investigate the impact of HL interventions on health outcomes and health promotion behaviors [36]. Further, follow-up in these studies was not systematic and longtime. It is therefore difficult to interpret whether the effect of HL interventions could improve health outcomes and health promotion behaviors in routine care. However, a trend showed that HL interventions might benefit low-social-economic groups more than the higher privileged population. This result was interpreted with caution and was not used in the meta-analysis. Nonetheless, we stratify the available interventional studies based on the outcomes to reduce potential bias. Likewise, the funnel plot was used to examine potential publication biases. This plot shows no explicit gaps and the symmetry of the plot was reasonable.

## 5. Conclusions

Despite all limitations, the first quantitative synthesis of data on the status of HL and its relationship with health promotion behaviors in the Iranian population has provided important insights. Our findings indicated the fact that limited HL is still common in the Iranian community. This review highlights the beneficial impact of HL interventions on health outcomes, health promotion behaviors, and patient–physician communication. According to the results, the future study agenda in this topic must include (1) a specific focus and attention on HL in the different population, (2) the development of a shared and clear definition for HL, (3) designing the comprehensive local screening instrument for HL, and (4) improvement of the HL intervention based on appropriate sample size, correct sampling and randomization, and longitudinal and systematic follow up. From a practice perspective, addressing these research gaps could be sufficient to improve the HL skills and develop a well-designed intervention based on HL strategies.

## Figures and Tables

**Figure 1 ijerph-18-04260-f001:**
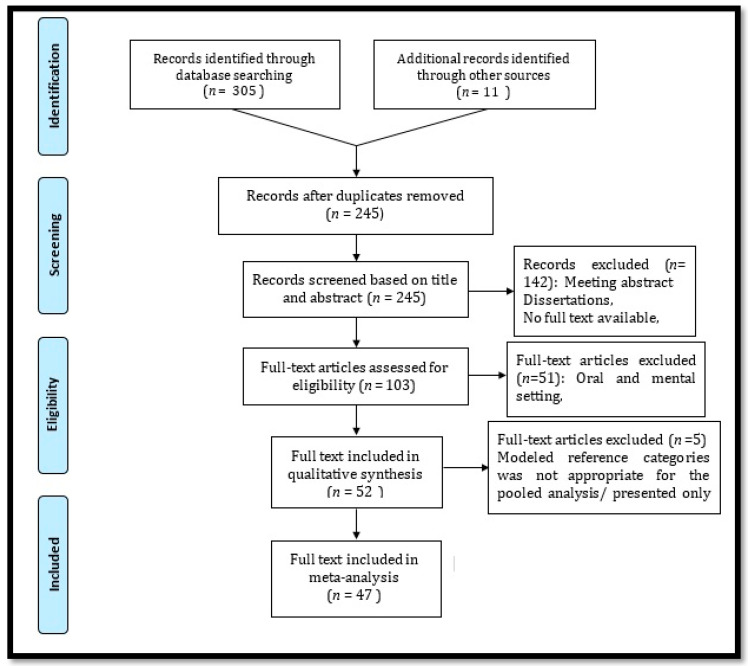
PRISMA flow diagram.

**Figure 2 ijerph-18-04260-f002:**
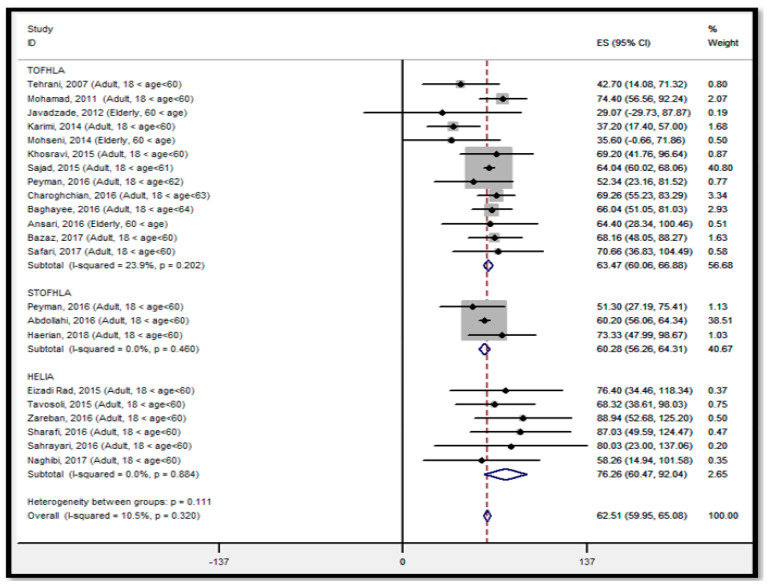
Forest plot of the distribution of effect size for health literacy status in populations with a health condition, stratified by type of instruments, ES: effect size, *I*^2^: a measure of the ratio of the total variation in study measures due to heterogeneity, *p* > 0.05: devastates heterogeneity is appropriate for interpretation.

**Figure 3 ijerph-18-04260-f003:**
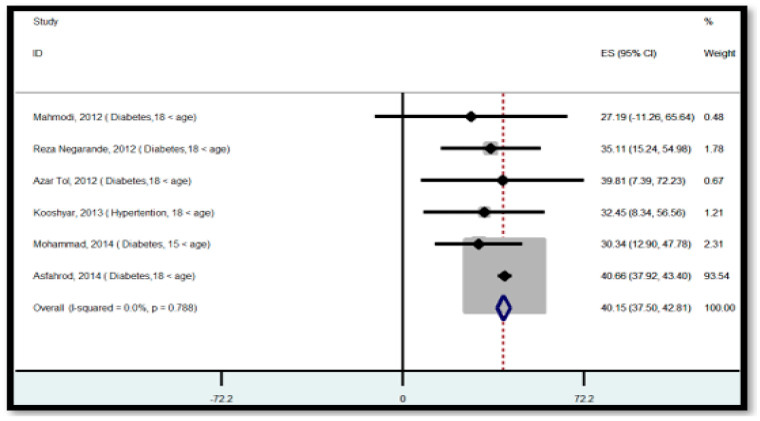
Forest plot of the distribution of effect size for health literacy status from 2012 to 2014, ES: effect size, *I*^2^: a measure of the ratio of the total variation in study measures due to heterogeneity, *p* > 0.05: devastates heterogeneity is appropriate for interpretation.

**Figure 4 ijerph-18-04260-f004:**
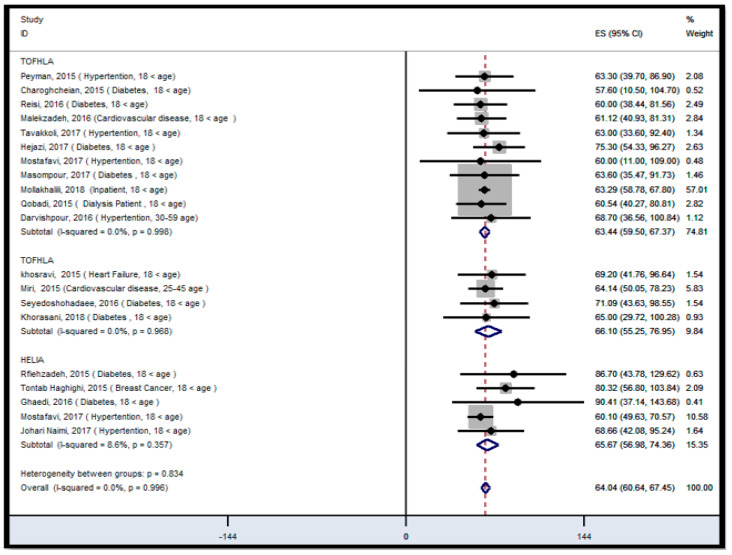
Forest plot of health literacy status in patient population after 2014, stratified by type of instruments, ES: effect size, *I*^2^: a measure of the ratio of the total variation in study measures due to heterogeneity, *p* > 0.05: devastates heterogeneity is appropriate for interpretation.

**Figure 5 ijerph-18-04260-f005:**
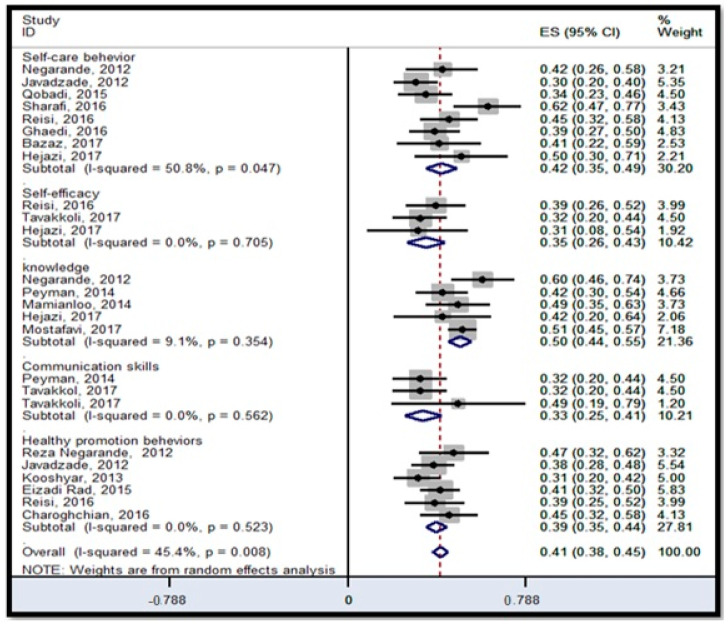
Forest plot for binary outcomes for the association between health literacy, self-care behaviors, self-efficacy, knowledge, communication skills, and health promotion behaviors, ES: effect size, *I*^2^: a measure of the ratio of the total variation in study measures due to heterogeneity, *p* > 0.05: devastates heterogeneity is appropriate for interpretation.

**Figure 6 ijerph-18-04260-f006:**
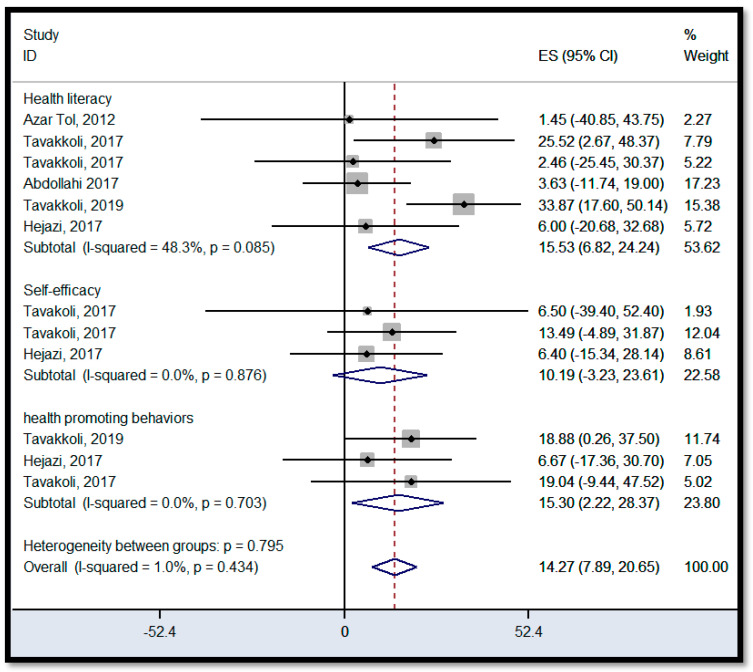
Forest plot for the intervention outcome. ES: effect size, *I*^2^: a measure of the ratio of the total variation in study measures due to heterogeneity, *p >* 0.05: devastates heterogeneity is appropriate for interpretation.

**Figure 7 ijerph-18-04260-f007:**
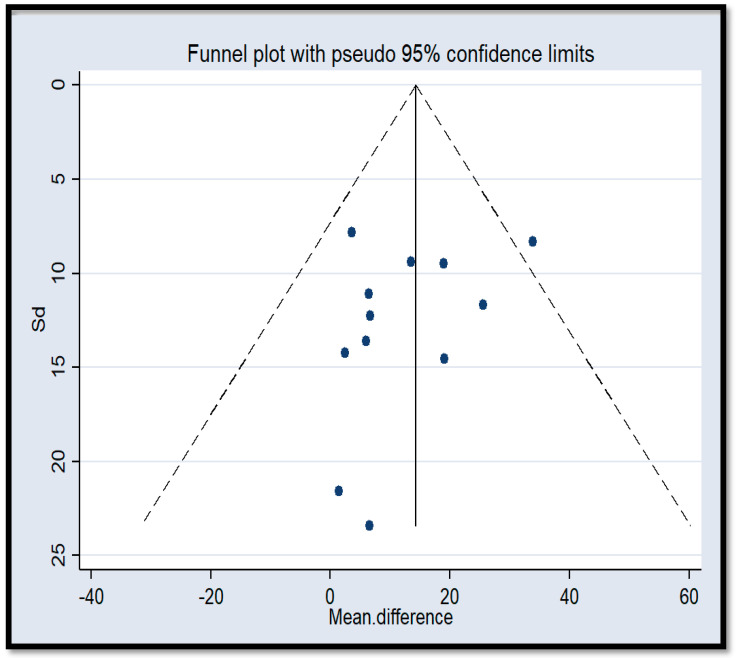
Funnel plot of standard error to assess publication bias for intervention outcomes, Sd: standard error.

**Table 1 ijerph-18-04260-t001:** Characteristics of included studies.

Authors (Year)	Tool	City	Study Design	N	Participants	Mean H	HL Scores (%)
Mahmodi, 2012 [41]	TOFHLA	Saqqez	Cross-sectional	1563	Diabetic patient, 65% illiterate, 35% diploma or higher	27.19± 19.62	Ad: 6.2, M: 14.8, In: 79
Negarande, 2012 [42]	TOFHLA	Tehran	RCT	127	Diabetic patient, 95% under diploma	35.11 ± 10.14	
Azar Tol, 2012 [43]	TOFHLA	Tehran	RCT	160	Diabetic patient, 98.1% married, 65% >diploma, 67.7% moderate or higher income	39.81 ± 16.54	
Mamianloo, 2014 [44]	NVS	Tehran	Semi-experimental	150	Heart failure patient, 41% diploma/40% under diploma, 58% married	1.77 ± 0.4	
Mohammad, 2014 [63]	TOFHLA	Tehran,	Cross-sectional	407	Diabetic patient, 71% illiterate, 35% under diploma	30.34 ± 8.9	Ad: 18.2, M: 17.8, In: 70
Reazee 2014 [45]	TOFHLA,	Yazd	Cross-sectional	432	Diabetic patient, 99.1% married, 85% moderate or high income, 69% under diploma, 7.4% live in village	46.66 ± 1.4	Ad: 22.2, M: 18.5, In: 59.3
Peyman, 2014 [46]	TOFHLA	Mashhad	Cross-sectional	240	Hypertensive patient, 75% illiterate or under diploma, 80% low income	63.3 ± 12.04	Ad: 50.4, M: 37.9, In: 11.7
Khosravi, 2015 [27]	TOFHLA	Bushehr	Cross-sectional	250	Heart failure patient, 75% diploma or under diploma,	69.2 ± 14	
Miri, 2015 [5]	TOFHLA	Mashhad	Cross-sectional	75	Cardiovascular disease, 82% illiterate or under diploma	64.14 ± 7.19	Ad: 14.66, M: 38.6, In: 38.67
Charoghcheian, 2015 [48]	TOFHLA	Chenaran	Semi-experimental	162	Diabetic patient, 92% under diploma, 68% low income	53.6 ± 24.03	Ad: 11, M: 30.5, In: 68.5
Reisi, 2016 [49]	TOFHLA	Isfahan	Cross-sectional	187	Diabetic patient, 57% under diploma, 95% married	60 ± 11,	
Malekzadeh, 2016 [50]	TOFHLA	Kerman	Cross-sectional	200	Cardiovascular disease, 68% diploma or under diploma, low or moderate income, 98% Married	61.12± 10.3	Ad: 29, M: 22, In: 49
Seyedoshohadaee, 2016 [30]	TOFHLA	Tehran	Cross-sectional	200	Diabetic patient, 90% married, 92% low or moderate income, 69 diploma or under diploma	71.09± 14.01	Ad: 42, M: 24, In: 24
Tavakkoli, 2017 [51]	TOFHLA	Mashhad	RCT, 2 interventions	240	Hypertensive patient, 82% Married, 80% under diploma, 79% low income. Two health providers	63 ± 15	Ad: 32, M: 28, In: 40
Hejazi, 2017 [52]	TOFHLA	Mashhad	Semi-experimental	70	Diabetic patient, 77% diploma or under diploma, 92% low or middle income	75.3 ± 10.7	Ad: 37, M: 35, In: 28
Mostafavi, 2017 [29]	TOFHLA	Esfahan	Cross-sectional	700	Hypertensive patient, 91% married, 72% under diploma	60 ± 25	
Masoompour, 2017 [53]	TOFHLA	Kerman	Cross-sectional	400	Diabetic patient, 81% married, 84% illiterate or under diploma	63.6 ± 14.35	
Khosravi 2018 [54]	TOFHLA	Shiraz	Cross-sectional	400	Diabetic patient, 75% diploma or under diploma	65 ± 18, 17–99	Ad: 41.4, M: 23.6, In: 35
Mollakhalili, 2014 [55]	TOFHLA	Isfahan	Cross-sectional	384	Inpatient, 59% diploma or under diploma, 96% low or middle income, 62% married	63.29± 2.3	Ad: 33.9, M: 25, In: 41
Kooshyar, 2013 [55]	STOFHLA	Mashhad	Cross-sectional	300	Hypertensive patient, 62% diploma or under diploma, 82% married	32.45± 12.3	Ad: 15, M: 14, In: 71
Qobadi, 2015 [56]	STOFHLA	Tehran	Cross-sectional	240	Dialysis Patient, 59% diploma or under diploma, 71% married	60.54 ± 10.34	Ad: 65.2, M: 9.8, In:25
Darvishpour, 2016 [57]	STOFHLA	Rasht	Cross-sectional	257	Hypertensive patient, 69% diploma or under diploma	68.7 ± 16.4	Ad: 41.62, M: 30, In: 28.4
Rfiehzadeh, 2015 [58]	HELIA	Gorgon	Cross-sectional	100	Diabetic patient, 54% diploma or under diploma, 89% married	86.7± 21.9	Limited: 79, In: 21
Haghighi, 2015 [59]	HELIA	Tehran	Cross-sectional	260	Women with breast cancer, 74% diploma or under diploma	80.32± 12	In: 6.9, Ad: 38.3 Limited: 18.8,
Ghaedi, 2016 [60]	HELIA	Bastak	Cross-sectional	265	Diabetic patient, 80% married, 83% illiterate or under diploma	90.41 ± 27.18	In: 51.7, Ad: 48.3
Johari Naimi, 2017 [61]	HELIA	Tehran	Cross-sectional	400	Hypertensive patient 73% diploma or under diploma,	68.66 ± 13.56	In: 7.8, limited: 55, Ad: 37.2
Tehrani, 2007 [9]	TOFHLA	5 Province	Cross-sectional	1086	41% illiterate/under diploma, 31% diploma 57% married, 45% low/moderate income, 35% live in village	42.7± 14.6	Ad: 28, M: 15.3, In: 56.6
Nekoei-Moghadam, 2011 [10]	TOFHLA	Kerman	Cross-sectional	1000	37% diploma/30% under diploma, 97% low or moderate income	74.4± 9.1	Ad: 41.4, M: 53.8, In: 4.8
Javadzade, 2012 [16]	TOFHLA	Esfahan	Cross-sectional	354	Elderly, 58% illiterate or under diploma/32% diploma, 77.7% narried, 86% low income	29.07 ± 30	Ad: 10.3, M: 23.7, In: 66
Azimi, 2013 [15]	NVS	Tehran	Cross-sectional	250	University student, 100% higher diploma, 90% single,	1.84 ± 1.36	In: 44.8, limited: 44.4, Ad: 10.8
Tavassoli, 2013 [62]	NVS	Esfahan	Cross-sectional	525	-	2.4± 1.45	In: 26.5, limited: 36.5, Ad: 38
Karimi, 2014 [17]	TOFHLA	Esfahan	Cross-sectional	300	41% under diploma/30% diploma, 82% married	37.2 ± 10.1	
Mohseni, 2014 [64]	TOFHLA	Kerman	Cross-sectional	200	Elderly, 87% illiterate/under diploma, 92% married	35.6 ± 18.5	Ad: 17, M: 31, In: 52.5
Sajjadi, 2015 [18]	TOFHLA	Izeh	Cross-sectional	240	Adult women, 42% diploma/50% under diploma, 100% married	64.04 ± 2.05	Ad: 38.75, M: 37.91, In: 23.33
Peyman, 2016 [19]	TOFHLA	Khaf	Cross-sectional	43	55% under diploma/40% diploma	52.34± 14.89	
Charoghchian, 2016 [20]	TOFHLA	Mashhad	Cross-sectional	185	Pregnant women, 33% diploma/40% under diploma, 100% married, 79% low income	69.26 ± 7.16	Ad: 35.10, M: 39.80, In: 15.1
Baghaei, 2016 [21]	TOFHLA	Orumiyeh	Cross-sectional	400	Pregnant women, 46% diploma /52% under diploma, 86% low income, 100% married	66.04 ± 7.65	Ad: 51, M: 25, In: 24
Ansari, 2016 [22]	TOFHLA	Zahedan	Cross-sectional	200	Elderly, 33% diploma/40% under diploma, 87% married, 78% low/moderate income	64.4 + 18.4	Ad: 32.5, M: 29, In: 38.5
Bazaz, 2017 [57]	TOFHLA	Ahvaz	Cross-sectional	93	Women population	68.16 ± 10.26	Ad: 46, M: 31.8, In: 22.2
Safari, 2017 [23]	TOFHLA	Bandar Abbas	Cross-sectional	250	Pregnant women, 24% diploma/32% under diploma, 100% married	70.66 ± 17.26	Ad: 52, M: 20.8, In: 27.2
Peyman, 2016 [24]	STOFHLA	Mashhad	Cross-sectional	120	Postpartum women, 26.2% diploma/40% under diploma, 60% low income, 100% married	51.3 ± 12.3	
Abdollahi, 2016 [25]	STOFHLA	Mashhad	Semi-experimental	80	Postpartum women, 80% diploma or higher, 60% low income,	60.2 ± 2.11	Ad: 25.3, M: 35.1, In: 39.6
Haerian, 2018	STOFHLA	Yazd	Cross-sectional	224	-	73.33 ± 12.93	
EizadiRad, 2015 [65]	HELIA	Baluchistan	Cross-sectional	400	26% diploma/43% under diploma, 81% low/moderate income	76.4 ± 21.4	Ad: 32, limited: 34, In: 34
Tavassoli, 2015 [26]	HELIA	Iran population	Cross-sectional	20571	-	68.3± 15.16	In: 12, limited: 32.4, Ad: 39.9, Ex: 15.8
Zareban, 2016 [28]	HELIA	Zahedan	Cross-sectional	247	Women, 24% diploma/40% under diploma, 85% low middle income, 100% married	88.9 ± 18.5	In: 33.2, limited: 34, Ad: 32
Sharafi, 2016 [29]	HELIA	Tehran	Cross-sectional	105	64% illiterate or under diploma/30% diploma 49.5% married	87.03 ± 19.1	
Sahrayi, 2016 [30]	HELIA	Karaj	Cross-sectional	525	32% diploma/30% under diploma, 53.5% married	80.03 ± 29.1	In: 24.2, limited: 23.4, Ad: 37.9, Ex: 14.5
Afshari, 2016 [31]	HELIA	Tehran	Cross-sectional	157	64% illiterate or under diploma/ 22% diploma, 100% low income	45.32 ± 19.3	In: 79, limited: 22.09, Ad: 20
Naghibi, 2017 [32]	HELIA	Shahryar	Cross-sectional	299	21.8% diploma/30.54% under diploma	58.26± 22.1	In: 36.5, limited: 23.1, Ad: 23.1, Ex: 14.4
Kaboudi, 2017 [33]	HELIA	Kermanshah	Cross-sectional	420	University student, 100% higher diploma, 95% single	40.04 ± 0.43	
Tavakkoli, 2019 [62]	TOFHLA	Kashmar	Semi-experimental	80	HF Heart failure patient, 28% diploma or 43% under diploma, 95% married	42.03 ± 5.37	Ad: 12.4, In: 87.6

N: Sample size, Ad: Adequate, M: Marginal, In: Inadequate, Ex: Excellent, RCT: randomized control trial.

## Data Availability

Authors should accurately present their research findings and include an objective discussion of the significance of their find-ings.

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
