# Peer review of "The Health Literacy Status and Its Role in Interventions in Iran: A Systematic and Meta-Analysis"

_ijerph, 2021, doi:10.3390/ijerph18084260_

Round 1

Reviewer 1 Report

Thank you for the opportunity to review the paper. Overall, I believe the methodological approach and scientific reasoning behind the paper are sound. The justification of the study is adequate, and I believe the paper fills some gaps in the literature. Methods are well-described and statistical techniques are sound.

My most significant concern is that English needs extensive editing throughout, I strongly recommend that the authors consider using professional English editing services. I had to read through the text several times before I was confident that I understood what the authors meant.

The authors need to further elaborate on the fact that many studies included for analysis are cross-sectional. Did the authors examine whether some of these studies needed to be excluded because of confounders? (or whether these confounders were adequately addressed?)

Why is self-efficacy a central feature of the mechanism of intervention in the authors' approach? This needs to be explained in the introduction section. 

Other minor comments include:

-Check table and figure headings and formatting

-Provide definitions of abbreviations used in the tables.

-The image resolution for the figures are too low

-The concluding sentence, "improve the low HL individuals' skills" - what skills? Please clarify here and in other parts of the document. 

Author Response

Reviewer reports 1

Dear Dr/ Prof,

We would like to thank the reviewer for careful reading of this manuscript and for the thoughtful comments and constructive suggestions, which help to improve the quality of this manuscript. We tried to answer to all valuable comments/suggestions/queries and all answers to comments are highlighted within the document by using Yellow color.

General comment: Thank you for the opportunity to review the paper. Overall, I believe the methodological approach and scientific reasoning behind the paper are sound. The justification of the study is adequate, and I believe the paper fills some gaps in the literature. Methods are well-described and statistical techniques are sound.

Comment 1: My most significant concern is that English needs extensive editing throughout, I strongly recommend that the authors consider using professional English editing services. I had to read through the text several times before I was confident that I understood what the authors meant.

Reply 1: Based on this comment, professional English editing services proofread this manuscript, and all typographical and grammatical errors were corrected (Red color)

Comment 2: The authors need to further elaborate on the fact that many studies included for analysis are cross-sectional. Did the authors examine whether some of these studies needed to be excluded because of confounders? (or whether these confounders were adequately addressed?)

Reply 2: Full-text articles were retrieved for eligibility assessment based on inclusion and exclusion criteria. In this study, 52 studies met our inclusion criteria, which included 45 cross-sectional studies [1-25], 3 randomized control trials (RCT)[26-28], and 4 semi-experimental studies [12, 17, 29, 30]. Then, independent dual rating was performed to consider and appraise the quality of included studies. Inconsistencies were resolved through moderated discussions. The quality of included studies was assessed using the Cochrane risk of bias tool [31], Downs and Black quality assessment scores[32], and JBI critical appraisal checklist [33] to examine the quality of randomized controlled trials (RCT), non-randomized intervention, and cross-sectional studies, respectively. The quality of included studies was low, across RCTs. Three studies were rated against the Cochrane risk of bias tool. All RCTs did not clearly report allocation concealment, blinding of participants, and blinding of outcome assessment while all included studies reported the random sequence generation, selective reporting, outcome assessment, and other sources of bias. The quality of included semi-experimental studies was acceptable with an average score of 13.6. It ranged from 12.9 to 14 out of 26. This is moderate quality but consistent with Downs and Black quality assessment scores for cross-sectional studies [32].

The quality of included studies was moderate, across cross-sectional studies (see Table S2). All cross-sectional studies were rated against the JBI critical appraisal checklist. Almost all of the studies clearly defined criteria for inclusion, study subject and the setting, and appropriate statistical analysis. All cross-sectional studies also used objective, standard criteria for measurement of the condition and provided a valid and reliable way for outcome measure. Five studies did not identify confounding factors and strategies to deal with confounding factors. Only three studies did not clearly define the way to measure exposure. Caution is needed in inferring these results and these data were not used in the meta-analysis process. Therefore, it is essential to conduct longitudinal studies to assess the effect of socio-demographic characteristics on HL status.

Table S2: Joanna Briggs Institute critical appraisal checklist for analytical cross-sectional studies

Critical appraisal checklist

Yes

No

Unclear

Not applicable

1

Were the criteria for inclusion in the sample clearly defined?

45

0

0

0

2

Was the exposure measure in valid and reliable way

42

0

3

0

3

Were objective, standard criteria used for measurement of the condition

45

0

0

0

4

Were confounding factors identified?

40

5

0

0

5

Were the study subject and the setting described in detail?

45

0

0

0

6

Were strategies to deal with confounding factor stated?

40

5

0

0

7

Was the outcome measured in a valid and reliable way?

45

0

0

0

8

Was appropriate statistical analysis used?

45

0

0

0

Comment 3: Why is self-efficacy a central feature of the mechanism of intervention in the authors' approach? This needs to be explained in the introduction section. 

Reply 3: The effect of self-efficacy as a central feature of the mechanism of intervention was not examined in this study because the main target of this study was to assess the HL status of the Iranian population and to examine whether HL interventions improved the HL skills, health-promoting behaviors, and self-efficacy. Further, the challenge, gaps, limitations and the need for future research are discussed.

Other Minor Comments Include:

Comment 4: Check table and figure headings and formatting: Done

Comment 5: Provide definitions of abbreviations used in the tables: Done

Comment 6:The image resolution for the figures are too low: Done

Comment 7: The concluding sentence, "improve the low HL individuals' skills" - what skills? Please clarify here and in other parts of the document. 

Reply 7: health literacy is the use of a wide range of skills that improve the ability of people to act on information in order to live healthier lives. These skills include reading, writing, listening, speaking, numeracy, and critical analysis, as well as communication and interaction skills.”

Reviewer 2 Report

The manuscript deals with an important issue but it is limited to studies in Iran. A broader scope would improve the manuscript but there may also be good argument for summarizing the Iranian studies.

Major comments:

  1. The concept of health literacy could be described in a more constructive way. It is nor convincingly wtitten with regard to health literacy as a risk factor and as an asset as well as mesurement of health literacy. For additional information see Don Nutbeim (2008). The evolving concept of health literacy. Social Science & Medicine 67, 2072–2078.
  2. There is a bit of confusion on the number of studies included in the paper. This need to more informatively descibed as noted below.

Minor comments:

  1. Line 25: Why do you report two decimals for all averages in the paper? It gives an impression of a very high precision.
  2. Line 41: The direct reference is to Simonds SK (1974). Health education as social policy. Health Education Monograph, 2:1-25.
  3. In the introduction it would be of interest to include some words on the use of health literacy in policy and health sector reforms, see Kristine Sørensen, Anita Trezona, Diane Levin-Zamir, Urska Kosir, Don Nutbeam (2019). Transforming health systems and societies by investing in health literacy policy and strategy. PUBLIC HEALTH PANORAMA 2–3,SEPTEMBER 2019 | 123–329.
  4. Line 60: These are just a limited selection of studies [7,8] more should be added covering a broader field of health literacy studies.
  5. Line 70: Why is this reference refered to?
  6. Line 70: Rewrite the aim of the study as you have two aims.
  7. Line 85; You use the "concept health promotion behaviors" where you include medical adherence. You can use the broader concept "health behavior" or you have to differentiate between health behavior and illnes behavior. I think it would be good to use the concepts used in medical or health sociology.. Moreover it should be "healthy food" not just "health food".
  8. Line 92: "Snowballing" technique gives an impression of not beeing systematic. It could be rewritten to better reflect the ambition of the researchers.
  9. Line 123: The STROBE measures need to be presented as it is used later, how did the0.63 on line185 came qabout.
  10. Line 135: The correlation coefficient was used. Does this means that a negative correlation was related to a negative effect not no effect?
  11. Line 181; Why do Table 1 include 26 studies while in the text there are 24 studies? Moreover, the references are given by name of the first author, which made it difficult to find in the list of references. Why do you nor include [number of reference] in the table?
  12. Line 183; The Prisma Flow diagram should include the number of studies in each step. Please add this information.
  13. Line 211: As is shown in Table 1 - this is not shown but I had to count the data. Masybe a line with the total for the whole group should be added to the table.
  14. Line 213: Now it is 28 studies?
  15. Line 230: 52 studies and 46 included. Please make a clear discription of the studies included in the study and the separate analysis.
  16. Line 234; "Metal analysis" shoud be "meta analysis". 
  17. Line 267: The legend to figure 2 must be correct, please rewrite and explain the scale used. Is it possible to have negative values. Is it the Effect Size a really proper concept to be used in this situation? It is the mean value in different groups.
  18. Line 270: The legend to figure 3 is not correct. It is not straztified by type of instrument. It is not reported which instruments are used. Moreover, the scale is not described. Please correct all legends to figures!
  19. Line 309: The spelling of Tavakoli is different Tavakoly and Tavakkoli?
  20. Line 318: Where has "60,68%" been reported before, new results should not be introduced in the discussion.
  21. Line 369; Is it nor circular reasoning to say that HL level has a dignificant association with people's self-care behavior and self-efficacy, knowledge, commmunication shills, and helath promotion behavior as these are components of HL?
  22. Line 391: The outlier? Are ther any other differences between the Tavakkoli's study and the other? Is the study correct for the type of pateinets included?
  23. Line 448: "Forth" should be "Fourth"
  24. LIne 461: two typographic errors need to de corrected.
  25. Line 482: Is it nor self-evident that after exclusion of studies there are more limited effects of design etc.?
  26. Line 528: Typografic errors to address.
  27. References should be possible to retrieve. Therefore additional information is needed for references no: 18, 24, 35, 49, 59, 68, 83. Which of the references are not published in English. These should be indicated by [title of report/paper].

Author Response

Reviewer reports 2

Dear Dr/ Prof,

We would like to thank the reviewer for the careful reading of this manuscript and for the thoughtful comments and constructive suggestions, which help to improve the quality of this manuscript. We tried to answer all valuable comments/suggestions/queries and all answers to comments are highlighted within the document by using Green color.

Major comments:

Comment1: The concept of health literacy could be described in a more constructive way. It is not convincingly written with regard to health literacy as a risk factor and as an asset as well as the measurement of health literacy. For additional information see Don Nutbeim (2008). The evolving concept of health literacy. Social Science & Medicine 67, 2072–2078.

Reply 1: based on this valuable comment, further information related to the concept of HL and its measurement was explained as follow:

Relevant evidence had shown that HL is a complex issue, and low HL can increase an individual's health status, health outcomes, and the risk of misinterpreting treatment instructions [4]. The World Health Organization (WHO) defining HL as follows: “Health literacy implies the achievement of a level of knowledge, personal skills and confidence to take action to improve personal and community health by changing personal lifestyles and living conditions” [3]. This definition confirms that the concept of HL is not simply a functional ability (the ability to read and write), it includes different skills that empower people to attend more fully in the community and to exert a higher degree of control over their health action and health decision-making.

The concept of HL has emerged from two distinctive roots that describe HL, respectively, as a personal ‘‘asset’’, or a clinical ‘‘risk’’. The clinical ‘‘risk’’ reflects recognition of the impact that low HL can have on the effectiveness of health service organization and clinical care. Improved the sensitivity of service organizations and clinicians can enhance the quality of patient–health provider communication, and improve access to health care services. This leads a health service organization better placed to provide patient education and management that will help to improve patient ability and health outcomes associated with successfully adhered to recommended clinical care. The concept of HL as an “asset” has evolved from roots in public health, health promotion, health education, and communication. This conceptualization is seen as a means to developing individuals’ skills and ability to exert greater control over their health actions (social, personal, and environmental) and health decision-making. Actions to improve HL are focused on the development of context intended to promote an individual’s health knowledge, self-efficacy, and self-management. This conceptualization of HL is less well tested through systematic research. Research to support the ‘‘asset’’ model is at a basic stage and it remains the main idea.

Likewise, if achieving HL as described by WHO is to be the main aim, different measurement tools will be needed for different stages and ages. Different measurement tools will be required to distinguish between interactive, critical, and functional HL. Although assessing the concept of HL is not totally new challenge in the social sciences, there will need comprehensive testing to ensure that the different measures not only focuses on health information for personal benefit but also on the social determinants of health.

Comment 2: There is a bit of confusion on the number of studies included in the paper. This needs to more informatively described as noted below.

Reply 2: A total of 52 potentially relevant articles with data on 36523 participants were included in the systematic review (Table 1). Of these, 47 studies with data on 34171 participants met meta-analysis inclusion criteria and included in the meta-analysis (Figure 1).

Minor comments:

Comment 3: Line 25: Why do you report two decimals for all averages in the paper? It gives an impression of very high precision.

Reply 3: in this manuscript, we used two decimals for all averages because gives an impression of precision. Some studies reported that this issue depends on the accuracy of the tools we employ in our research, each variable is measured within a certain degree of precision. However, apparently, there is no consensus on this issue. For example, some references suggest that in reporting statistics (eg, means and standard deviations [SDs] not to use precisions higher than the accuracy of the measured data; many researchers recommend using only one or two decimal places more than the precision used to measure the variable; and, some mention that although means should not be reported to any more than one decimal place more than that of the raw data, SDs may need to be reported with an extra decimal place. Therefore, in this manuscript, based on the

Comment 4: Line 41: The direct reference is to Simonds SK (1974). Health education as social policy. Health Education Monograph, 2:1-25.

Reply 4: reference was changed

Comment 5: In the introduction, it would be of interest to include some words on the use of health literacy in policy and health sector reforms, see Kristine Sørensen, Anita Trezona, Diane Levin-Zamir, Urska Kosir, Don Nutbeam (2019). Transforming health systems and societies by investing in health literacy policy and strategy. PUBLIC HEALTH PANORAMA 2–3, SEPTEMBER 2019 | 123–329.

Reply 5: relevant information to health literacy in policy and health sector reforms was explained in introduction section as follow: Globally, Hl policies and strategies are not yet a well known among health decision-maker and politicians. According to WHO, today’s reaction-based health care centers are not suitable for the challenges of the recent century because half the world’s population access to poor quality health care services. Health systems transformation around the needs of communities and people is more effective to improve HL and patient engagement[34].

  1.  

Comment 7: Line 60: These are just a limited selection of studies [7,8] more should be added covering a broader field of health literacy studies.

Reply 7: broader field of HL studies added

Comment 8: Line 70: Why is this reference referred to? Deleted

Comment 9: Line 70: Rewrite the aim of the study as you have two aims. Done

Comment 10: Line 85; You use the "concept health promotion behaviors" where you include medical adherence. You can use the broader concept "health behavior" or you have to differentiate between health behavior and illness behavior. I think it would be good to use the concepts used in medical or health sociology.. Moreover, it should be "healthy food" not just "health food".

Reply 10: In the studies that met the inclusion criteria of this review, only health nutrition behaviors, physical activity behaviors, and medication adherence were examined as health behaviors. However, based on this comment, we differentiate between health behavior and illness behavior.

Comment 11: Line 123: The STROBE measures need to be presented as it is used later, how did the0.63 on line185 come about.

In this review, we used strobe for the basic assessment of cross-sectional studies. Since,

Strobe statement is a qualitative tool, not a quantitative score (There is no score for the Strobe checklist), we used the JBI critical appraisal checklist to assess cross-sectional studies. So, quality control of research publication was rewritten in manuscript based on further details as follow:

Full-text articles were retrieved for eligibility assessment based on inclusion and exclusion criteria. In this study, 52 studies met our inclusion criteria, which included 45 cross-sectional studies [1-25], 3 randomized control trials (RCT)[26-28], and 4 semi-experimental studies [12, 17, 29, 30]. Then, an independent dual rating was performed to consider and appraise the quality of included studies. Inconsistencies were resolved through moderated discussions. The quality of included studies was assessed using the Cochrane risk of bias tool [31], Downs and Black quality assessment scores[32], and JBI critical appraisal checklist [33] to examine the quality of randomized controlled trials (RCT), non-randomized intervention, and cross-sectional studies, respectively. (Please see section: 2.3. Methodological quality and Data Extraction).

The quality of included studies was low, across RCTs. Three studies were rated against the Cochrane risk of bias tool. All RCTs did not clearly report allocation concealment, blinding of participants, and blinding of outcome assessment while all included studies reported the random sequence generation, selective reporting, outcome assessment, and other sources of bias. The quality of included semi-experimental studies was acceptable with an average score of 13.6. It ranged from 12.9 to 14 out of 26. This is moderate quality but consistent with Downs and Black quality assessment scores for cross-sectional studies [32]. The quality of included studies was moderate, across cross-sectional studies (see Table S2). All cross-sectional studies were rated against the JBI critical appraisal checklist. Almost all of the studies clearly defined criteria for inclusion, study subject and the setting, and appropriate statistical analysis. All cross-sectional studies also used objective, standard criteria for measurement of the condition and provided a valid and reliable way for outcome measure. Five studies did not identify confounding factors and strategies to deal with confounding factors. Only three studies did not clearly define the way to measure exposure. Caution is needed in inferring these results and these data were not used in the meta-analysis process. (Please see Methodological quality).

Table S2: [35]for analytical cross sectional studies

Critical appraisal checklist

Yes

No

Unclear

Not applicable

1

Were the criteria for inclusion in the sample clearly defined?

45

0

0

0

2

Was the exposure measure in valid and reliable way

42

0

3

0

3

Were objective, standard criteria used for measurement of the condition

45

0

0

0

4

Were confounding factors identified?

40

5

0

0

5

Were the study subject and the setting described in detail?

45

0

0

0

6

Were strategies to deal with confounding factors stated?

40

5

0

0

7

Was the outcome measured in a valid and reliable way?

45

0

0

0

8

Was appropriate statistical analysis used?

45

0

0

0

Comment 12: Line 135: The correlation coefficient was used. Does this mean that a negative correlation was related to a negative effect, not no effect?

Reply 12: A negative “r” shows that HL is inversely associated with better health outcomes; a positive “r” indicates that HL is associated.

Comment 13: Line 181; Why does Table 1 include 26 studies while in the text there are 24 studies?

Reply 14: Table 1 included 52 studies that met inclusion criteria for the systematic review. Of these, 47 studies with data on 34171 participants were included in the meta-analysis. Of 52 included studies, 25 (48%) studies included populations with health condition [1-25], and 24 (46.3%) studies included patient participants with chronic disease [4, 19, 26-30, 36-52], and other disease (3/52; 5.7%)[46, 47, 50] (Table1).

Comment 14: Moreover, the references are given by name of the first author, which made it difficult to find in the list of references. Why do you not include [number of references] in the table?

Reply 14: Reference add into Table 1

Comment 15: Line 183; The Prisma Flow diagram should include the number of studies in each step. Please add this information. Done

Comment 16: Line 211: As is shown in Table 1 - this is not shown but I had to count the data. Maybe a line with the total for the whole group should be added to the table.

Reply 16: added

Comment 17: Line 213: Now it is 28 studies? Corrected

Comment 18: Line 234; "Metal analysis" should be "meta-analysis". Done

Comment 19: Line 267: The legend to figure 2 must be correct, please rewrite and explain the scale used. Done

Comment 20: Line 270: The legend to figure 3 is not correct. It is not stratified by type of instrument. It is not reported which instruments are used. Moreover, the scale is not described. Please correct all legends to figures!. Done

Comment 21: Line 318: Where has "60,68%" been reported before, new results should not be introduced in the discussion. Corrected

Comment 22: Line 369; Is it not circular reasoning to say that HL level has a significant association with people's self-care behavior and self-efficacy, knowledge, communication skills, and health promotion behavior as these are components of HL?

Reply 22: Yes, it could be circular reasoning. However, we indicated this statement only based on results in an included study in the meta-analysis.

Comment 23: Line 391: The outlier? Are there any other differences between the Tavakkoli's study and the other? Is the study correct for the type of patients included?

Reply 23: No, there are no differences between the Tavakkoli's study and the other.

It certainly fits the trend observed in the other outcome.

Comment 24: Line 448: "Forth" should be "Fourth": Corrected

Comment 25: Line 461: two typographic errors need to de corrected. Corrected

Comment 26: Line 482: Is it not self-evident that after exclusion of studies there are more limited effects of design etc.?

Reply 26 this sentence was deleted from the manuscript.

Comment 27: Line 528: Typographic errors to address. Corrected

Comment 28: References should be possible to retrieve. Therefore additional information is needed for references no: 18, 24, 35, 49, 59, 68, 83. Which of the references are not published in English. These should be indicated by [title of report/paper]. Done

Round 2

Reviewer 1 Report

The authors have made a good faith effort in their revision. 

Reviewer 2 Report

The manuscript has been considerably improved and the present version can be published. However, there are still minor errors in the english language that can be corrected before publication of this interesting manuscript.